# De Novo Assembly-Based Analysis of *RPGR* Exon ORF15 in an Indigenous African Cohort Overcomes Limitations of a Standard Next-Generation Sequencing (NGS) Data Analysis Pipeline

**DOI:** 10.3390/genes11070800

**Published:** 2020-07-15

**Authors:** Jordi Maggi, Lisa Roberts, Samuel Koller, George Rebello, Wolfgang Berger, Rajkumar Ramesar

**Affiliations:** 1Institute of Medical Molecular Genetic, University of Zurich, 8952 Schlieren, Switzerland; maggi@medmolgen.uzh.ch (J.M.); koller@medmolgen.uzh.ch (S.K.); 2University of Cape Town/MRC Genomic and Precision Medicine Research Unit, Division of Human Genetics, Department of Pathology, Institute of Infectious Disease and Molecular Medicine (IDM), Faculty of Health Sciences, University of Cape Town, Cape Town 7925, South Africa; lisa.roberts@uct.ac.za (L.R.); george.rebello@uct.ac.za (G.R.); raj.ramesar@uct.ac.za (R.R.); 3Zurich Center for Integrative Human Physiology (ZIHP), University of Zurich, 8006 Zurich, Switzerland; 4Neuroscience Center Zurich (ZNZ), University and ETH Zurich, 8006 Zurich, Switzerland

**Keywords:** RPGR, ORF15, RP, de novo assembly, diagnostics, genetic testing, secondary analysis

## Abstract

*RPGR* exon ORF15 variants are one of the most frequent causes for inherited retinal disorders (IRDs), in particular retinitis pigmentosa. The low sequence complexity of this mutation hotspot makes it prone to indels and challenging for sequence data analysis. Whole-exome sequencing generally fails to provide adequate coverage in this region. Therefore, complementary methods are needed to avoid false positives as well as negative results. In this study, next-generation sequencing (NGS) was used to sequence long-range PCR amplicons for an IRD cohort of African ancestry. By developing a novel secondary analysis pipeline based on de novo assembly, we were able to avoid the miscalling of variants generated by standard NGS analysis tools. We identified pathogenic variants in 11 patients (13% of the cohort), two of which have not been reported previously. We provide a novel and alternative end-to-end secondary analysis pipeline for targeted NGS of ORF15 that is less prone to false positive and negative variant calls.

## 1. Introduction

Retinitis pigmentosa (RP) is an inherited retinal disorder (IRD) that is characterised by the loss of retinal rod photoreceptor cells [1]. The initial symptom is usually nightblindness, followed by progressive constriction of the peripheral visual fields, leading to tunnel vision [2]. As the disease progresses, cone photoreceptor cells are inevitably affected, which can result in total blindness. RP is the most common IRD, with an estimated prevalence of about 1 in 3500–4000 individuals [1,3], and it can be inherited as an autosomal dominant (*ad*, 15–25%), autosomal recessive (*ar*, 35–50%) or an X-linked (*xl*, 7–15%) trait [3]. Simplex or isolated cases are also common [3,4].

A genetic diagnosis for IRD patients can confirm a clinical diagnosis, provide a clearer prognosis, form the basis of genetic counselling for families, and allow participation in gene-based therapy trials [5]. However, it is impossible to predict the causative RP gene based solely on phenotype and inheritance pattern. RP is usually monogenic, yet an expansive list of genes has been associated with the disease (RetNet^TM^, https://sph.uth.edu/retnet/). In addition to the exceptional genetic heterogeneity, phenotypic variability also occurs, complicating molecular analysis even further. However, it can be generalised that xlRP cases display some of the most severe RP phenotypes in males, with early age of onset (childhood to late teen years) and rapid disease progression [4,6]. To date, two major causative xlRP genes have been identified, *RP2* and *RPGR,* and they account for 10–20% and 50–90% of cases, respectively [4,6,7,8,9]. In addition, a deep intronic mutation in the syndromic ciliopathy gene, *OFD1*, has been identified in one family with severe non-syndromic xlRP [10].

The systematic screening of all RP genes in a cohort of patients using a candidate approach is challenging and costly due to the marked phenotypic and genetic heterogeneity described. Thus, next-generation sequencing (NGS) approaches such as whole exome sequencing (WES) or targeted resequencing are revolutionising molecular diagnostics in heterogeneous Mendelian diseases such as IRD, and specifically RP [11,12]. Still, accurate clinical information such as individual disease status, phenotypes, and mode of inheritance within a family are crucial. With ambiguities in these data, downstream analysis and variant prioritisation for mutation identification is impeded. For example, whilst males are exclusively affected in most X-linked recessive Mendelian disorders, female carriers of xlRP gene mutations may manifest with a spectrum of disease [3,6,13,14,15,16,17,18,19,20,21], which can result in a pedigree displaying an apparent *ad* trait. The incorrect assumption of disease inheritance (thereby, interrogating autosomal rather than *xl* variants) may cause false negative results.

One limitation of capture-based NGS is that a critical xlRP mutational hotspot is inadequately captured. The *RPGR* gene was identified as causing xlRP in 1996. Four years later, an RPGR transcript with a novel 3′ terminal exon, known as exon open reading frame 15 (ORF15), was identified [22]. ORF15 was reported as a mutational hotspot, accounting for the majority of xlRP cases studied. Multiple isoforms of RPGR exist, and the isoform containing ORF15 is the predominant transcript expressed in the retina [22,23]. ORF15 includes exon 15 and a portion of intron 15, and it is comprised of a highly repetitive, low-complexity, purine-rich sequence. The nature of this sequence has been postulated to be responsible for the region’s high mutability, possibly by adopting unusual structural conformations, thereby reducing the replication fidelity [22]. Additionally, this genomic sequence generates technical challenges in molecular investigations. It has been shown that the *RPGR* exon ORF15 is difficult to sequence, using conventional Sanger sequencing as well as NGS approaches, due to the highly repetitive sequence exceeding 1 Kb in length [24]. Huang et al. showed that ORF15 was insufficiently captured during NGS, causing false negative results. Therefore, they proposed that a complementary approach be taken in the molecular diagnosis of IRDs, with additional sequencing of targets such as *RPGR* ORF15, which evade capture-based technologies [24].

In an effort to elucidate the genetic landscape of IRD among the indigenous African populations of South Africa (SA), WES was previously performed in a cohort of 16 families, and a molecular diagnosis obtained for approximately 40% through an analysis of all reported IRD genes at that time (*n* = 285) [25]. Leveraging the vast genomic diversity of indigenous Africans in SA [26,27,28,29,30,31] yielded several novel mutations [25] and assisted the identification of a novel IRD gene [32]. Subsequently, the WES data from five unresolved families with an absence of male-to-male disease transmission were re-interrogated, confirming that ORF15 had been insufficiently captured. For 4/5 families, affected males within a family shared a haplotype spanning the region of interest. The genetic testing of *RPGR* exon ORF15 in those four probands identified two mutations in three families: c.2790_2791delGG; p.Glu931GlyfsTer147 and c.2964_2965delGG; and p.Glu989GlyfsTer89 (Appendix A). Both mutations had been previously associated with xlRP [33,34]. Therefore, supplementary analysis of ORF15 had resulted in identification of the probable causal mutation in 3/16 families (19%). The high detection rate premised our current study of the mutation hotspot in a larger cohort of indigenous African IRD patients as a pre-screen, prior to embarking on further WES.

NGS data analysis workflow can be divided into three stages of analysis: primary, secondary, and tertiary [35]. Briefly, primary analysis converts raw signals (i.e., images) into nucleotides sequences (FASTQ file generation). Secondary analysis aligns the sequences to a reference build (BAM file generation) or assembles a sequence de novo (contig file generation) and performs variant detection (VCF file generation). Finally, tertiary analysis involves the annotation of the resulting data and their interpretation. We report here a novel secondary analysis pipeline based on de novo assembly, which identified pathogenic variants in 11 patients (13% of the African cohort).

## 2. Materials and Methods

### 2.1. Patient Cohort

The University of Cape Town (UCT) IRD registry currently contains biological material, clinical information, and demographic data of 3506 individuals from 1617 SA Families with IRD. Informed consent was obtained according to the 2013 Declaration of Helsinki for all participants from whom samples have been archived. Ethics approval was granted by the Human Research Ethics Committee of the UCT Faculty of Health Sciences (REC REF: 226/2010 and 768/2013). Samples from the registry have been investigated over the years in various research projects, and only genetically unresolved indigenous African IRD cases were included in the present study. All individuals with pedigrees displaying male-to-male transmission of disease were excluded. The cohort comprised isolated males with RP (*n* = 65), probands with a family history suggestive of adRP, arRP, or xlRP (*n* = 17) and probands with recessive or sporadic macular degeneration (*n* = 15).

### 2.2. DNA Extraction

The UCT IRD registry and biobank has archived biological material from patients for three decades; therefore, various DNA extraction methods have been used over the years. Blood samples were collected in EDTA tubes and processed immediately, mainly using a salting out method [36], although different commercial DNA extraction kits have been trialled for short periods at various times. Alternatively, the buffy coat was removed from the blood sample and frozen, and it was later processed using the salting out method. Since 2013, saliva samples have been collected using Oragene^®^ collection kits (DNA Genotek Inc, Ottawa, Canada) and processed according to the manufacturers’ instructions.

DNA samples were quantified using the Nanodrop ND-1000 Spectrophotometer (Thermo Fisher Scientific, Waltham, MA, USA) and subsequently diluted to a final concentration of 20 ng/µL in Sabax water (Adcock Ingram, Johannesburg, SA). After dilution, the DNA integrity was confirmed using 1–2% weight/volume (*w*/*v*) agarose gels containing 0.5–1 g agarose (Lonza SeaKem^®^ LE, Basel, Switzerland), 50 mL of 1X Tris Borate EDTA (TBE) buffer and 5 µL SYBR^®^ Safe DNA Gel Stain (Thermo Fisher Scientific).

### 2.3. ORF15 NGS Amplification and Sequencing

The *RPGR* ORF15 was amplified and sequenced as previously described [37]. Briefly, primers were used to amplify a 2.1 kb region covering the entire exon ORF15 (hg19 chrX:38144633-38146732). Then, the amplicons were sheared in a microTUBE-50 AFA with a Covaris M220 instrument (Covaris, USA) to a mean size of 273 bp with the following settings: 75 W peak incident power, 10% duty factor, 200 cycles per burst, 170 s treatment time. Indexed DNA libraries were constructed with a TruSeq DNA Nano kit according to the manufacturer’s protocol (Illumina, San Diego, CA, USA). After end repair, size selection, A-tailing, indexed adapter ligation, and a last amplification step, the libraries were validated on a Bioanalyzer 2100 (Agilent, Santa Clara, CA, USA) and quantified with the Qubit dsDNA High sensitivity kit (Thermo Fisher Scientific). Final libraries were diluted to 4 nM for denaturation and prepared to a 12 pM solution for paired-end sequencing (2 × 151 cycles) on a MiSeq instrument with a MiSeq Reagent V2 (300-cycles) cartridge (Illumina).

### 2.4. NGS Data Analysis

Demultiplexed data were aligned to the Human reference genome hg19 by BWA-MEM (Burrows-Wheeler aligner), and variant calling was performed by GATK (Genome Analysis ToolKit). Additionally to this standard secondary analysis pipeline, FASTQ files also underwent de novo assembly with SPAdes (St. Petersburg genome assembler) [38]. The resulting contigs were filtered based on G-content and then scored based on size and coverage. If no appropriate contig was found (G-content <15%, size >2000 and <2300 bp, and coverage >30), the highest scoring contigs, together with the *RPGR* reference sequence, were given as trusted contig inputs in a second SPAdes assembly run with different kMER settings. The resulting contigs were scored again and selected. If still no appropriate contig was found, a third and final SPAdes assembly run was performed to resolve the last samples with kMERs 77, 99, and 127, the highest scoring contigs generated in the previous assembly steps as untrusted contigs, and the *RPGR* reference sequence as trusted contig inputs.

Then, the final contigs were aligned to the reference *RPGR* ORF15 sequence with MAFFT v7 (Multiple Alignment using Fast Fourier Transform) and screened for variants using SNP-sites for single-nucleotide variants (SNVs) and an in-house Python 3 (https://www.python.org) script for indels [39,40]. Finally, detected variants were annotated for frequency (gnomAD) [41]. The entire de novo assembly-based pipeline from FASTQ to gnomAD-annotated VCF is included in the Appendix A, as a JupyterLab notebook (https://jupyterlab.readthedocs.io/en/stable/). Patient NGS data are not available, due to privacy or ethical restrictions. The vast majority of reported pathogenic mutations in ORF15 (i.e., 257/259 variants in the Human Gene Mutation Database, HGMD^®^, and all variants in a gene-specific variant database) are frameshift or nonsense variants [42,43,44]. While it is possible that silent or missense variants may be pathogenic, these require extensive segregation analyses and functional studies to prove causality. Therefore, in this study, we prioritised only nonsense and frameshift variants.

### 2.5. Confirmation by Sanger Sequencing

Likely pathogenic variants were confirmed by Sanger sequencing and cosegregation analysis was performed in available familial samples. Moreover, assumed false positives and negatives generated by the standard pipeline were verified by Sanger sequencing when possible. A 1953 bp fragment spanning the coding region of ORF15 was amplified from 200 ng genomic DNA, using previously published primers ORF15-F3 and ORF15-R6 [45]. PCR was performed with the HotStarTaq^®^ kit (QIAGEN, Hilden, Germany), using the following cycling conditions: 95 °C—15 min, 45 cycles of {94 °C—45 s, 60 °C—1 min, 72 °C—3.5 min}, 72 °C—10 min, 10 °C hold.

Following purification with FastAP™ Thermosensitive Alkaline Phosphatase (Thermo Fisher Scientific) and Exonuclease I (Thermo Fisher Scientific), sequencing was performed with published primers ORF15-R7b, -R8b, -R8c, or -R9 [45] (depending on the location of the putative mutation), using BigDye^®^ Terminator v3.1 Ready Reaction Mix (Applied Biosystems by Thermo Fisher Scientific) and Q-Solution (QIAGEN).

Cycle sequencing products were purified by ethanol precipitation, and 5 µL purified sequencing reaction was added to 8 µL Hi-Di™ Formamide (Applied Biosystems by Thermo Fisher Scientific). Capillary electrophoresis was performed on an ABI Prism 3130xl Genetic Analyser (Applied Biosystems by Thermo Fisher Scientific). Output.ab1 files were aligned to the reference ORF15 sequence (NM_001034853.2) using Bioedit [46] and manually inspected for the specific variants identified by NGS, utilising Mutalyzer [47] to confirm the flanking sequences.

## 3. Results

### 3.1. NGS PCR and Sequencing Efficiency

Amplification of the ORF15 region of RPGR was possible for 82 of the 97 samples included in this study (85%). It was not possible to associate sample characteristics (such as extraction method, sample type, or sample age) to unsuccessful amplification. It is possible that variants in the primer-binding regions prevented amplification, at least in a subset of these samples.

The average DNA concentration after PCR for the successful samples was 35.1 ng/μL (range 5.4–132.0 ng/μL). The average maximum read depth in the ORF15 region after BWA-MEM alignment equalled 5346 reads.

### 3.2. De Novo Assembly Improves Diagnostic Sensitivity

RPGR’s ORF15 repeatedly proved itself not only challenging in terms of sequencing data generation, but also during subsequent data analysis. The standard NGS pipeline, consisting of reads alignment to the reference genome with BWA-MEM followed by variant calling with GATK, is not sufficient for this low-complexity region. Specifically, it often resulted in erroneous “heterozygous” calls in males, and it was particularly poor at detecting larger indels (≥20 bp). In fact, there were 606 total and 43 unique variants called across all sequenced samples that passed all quality filters; of the 606 variants, 546 were called as heterozygous, and of these, 432 (79%) were found in male samples. Furthermore, the standard analysis pipeline produced both false positives and false negative results.

To overcome this obstacle, we created an alternative Python-based pipeline (available in the Appendix A) integrating the de novo assembly software SPAdes, the multi-fasta aligner MAFFT v7, and the SNVs’ identified SNP sites [38,39,40]. To our knowledge, this is the first report of a de novo assembler being used for molecular diagnostic applications for inherited diseases. Briefly, SPAdes’ output is a FASTA file containing several (sometimes many) contigs. These contigs were scored based on size (<2300 and >400) and coverage as (>30) a pre-selection step. The reference sequence (plus strand) of the amplicon has less than 7% G-content. This characteristic was used to filter out unrelated contigs. In addition, MAFFT does not reverse complement; therefore, it is necessary that the selected contigs are from the same strand as the input reference sequence. For this reason, if the contig does not fulfill the G-content threshold, its reverse complement is checked for the same criterion. A satisfactory contig (size <2300 and >2040) was assembled and selected for 46/82 samples (56%). The other FASTQ files underwent a second round of assembly with SPAdes using the pre-selected top-scoring contigs (maximum 5 contigs) and the RPGR reference sequence as trusted contigs and different kMER settings. The second assembly step resulted in a final contig for 35 of the 36 remaining samples (97%). A third and final SPAdes assembly step was performed to resolve the last sample. When sequencing PCR amplicons with NGS, the primer-binding regions are generally over-represented compared to the rest of the amplicon; that causes some of the contigs to be larger than expected (up to 2258 bp in this cohort).

Then, the final contigs were aligned back to the reference RPGR sequence of the 2.1 kb amplicon using MAFFT v7 [39]. SNVs were called by SNP sites and indels were called by an in-house Python script [40]. Finally, all variants were annotated for their gnomAD frequency [41].

The Python pipeline allowed for the identification of indels in the low-complexity region that were otherwise missed by the standard NGS pipeline. Table 1 summarises the alternative pipeline findings for our cohort and shows whether the variant had been identified by the standard pipeline and whether it was confirmed by Sanger sequencing. The analysis revealed pathogenic variants in 11 patients (13.4%), including three recurrent mutations and two previously unreported frameshift mutations [7,22,33,34,48].

The identified indels could be confirmed in at least a subset of carriers. The novel pipeline called the c.2660_2661insGGAAGAGGAGGAAGGAGAAGGGGAGGGAGAAGAGGAAGGAGAAGGGGAGGG, c.2639A>G and c.2633_2634delinsAA variants in sample Afr-61; however, Sanger sequencing revealed that sample Afr-61 actually carries the c.2606_2632dup and an additional variant close-by: c.2670_2693dup (gnomAD (%) = 0.0016). The underlying sequence for either variant combination is the same; these are merely two different ways to call the variants compared to the reference sequence.

In contrast, Table 2 displays all variants passing all quality filters identified by the standard pipeline. It highlights how often each distinct variant has been called in the cohort by either secondary analysis pipeline. There is a remarkable discordance between the pipelines’ outputs. Each variant on Table 2 was assigned a call quality (either high or low) based on accordance with the novel pipeline’s results. High-quality calls were assigned to variants with complete concordance with the novel pipeline’s results.

Assuming all low-quality calls are false positive, the standard pipeline resulted in 516 false positive calls, whilst the novel pipeline produced 65 false positives. Unfortunately, it was not possible to test all discrepancies by Sanger sequencing. Instead, at least one carrier sample per discrepant variant was selected for Sanger sequencing to confirm the presence or absence of the respective variant.

Variants c.3074T>G, c.3062T>G, c.2863T>G, c.2758A>G, and c.2499T>G were called in most samples by the standard pipeline (in 80, 74, 81, 75, and 65 samples, respectively), but they were found only once by the novel pipeline (except for c.2863T>G, which was called in 47 samples and c.2758A>G in 4 samples). Moreover, the standard pipeline resulted in heterozygous calls exclusively, which we considered a low-quality variant marker. Sanger sequencing revealed these variants to be false positives in the tested subset (8 samples).

Variants c.2829T>G and c.2829T>A show a similar pattern; they were called in 50 and 30 samples, respectively, from the standard pipeline and only once by the novel pipeline. Both variants were confirmed to be false positive in 4 and 3 samples, respectively. Interestingly, these variants overlap with the c.2820_2840dup variant that has been identified by the novel pipeline in a subset of the corresponding samples. Even more striking is the overlap of samples carrying the c.2639A>G, c.2634G>A, c.2633G>A, and c.2589A>G variants in the standard pipeline’s outputs with those carrying the c.2606_2632dup variant in the novel pipeline’s output. All of the former variants were confirmed as false positives in a subset of the carriers, and the latter was confirmed as being present. These findings highlight that some of the standard pipeline’s false positive cluster around the region, where the novel pipeline identified indels.

All remaining low-quality call variants were not found in the novel pipeline’s output (except for c.2569A>G, which was called in 6 samples), and all those that could be verified by Sanger sequencing were confirmed to be false positives.

In contrast, all high-quality call variants tested could be confirmed, except for 2 variants from sample Afr-31, which is one out of five samples with an average coverage lower than 100×. It is noteworthy that most variants identified in these low-coverage samples are confirmed false positives (except for c.125delA, c.2606_2632dup, which could not be tested in these samples), suggesting that very deep coverage might be needed to result in high-quality variant calls with the novel pipeline.

Variants c.3407G>A, c.3396C>T, c.3264G>A, c.3219C>T, and c.2667_2669del were identified in one additional sample by standard analysis compared to the novel pipeline outputs; all missing samples were females called as heterozygous by the standard pipeline. These results seem to suggest that the novel pipeline might produce false negatives in female carriers. However, other heterozygous variants in female carriers were detected by the novel pipeline (c.2341G>A, c.2223G>A, c.1754-103C>T).

Figure 1 visually summarises the findings contained in Table 1 and Table 2 by illustrating the numbers of overall and unique true and false positives identified by each pipeline. It is important to point out once more that each unique variant call, which is present in multiple samples, has been Sanger verified in a subset of those. However, since all of them are concordant, all such variant calls have been considered as “confirmed” false or true positives in Figure 1.

### 3.3. Segregation Analysis

Eight pathogenic variants were confirmed in 11 families by Sanger sequencing (Figure 2). The lack of familial DNA samples was a limitation in this study, as is often the case in human genetic studies. Additional challenges were (a) amplification failure for some samples, possibly due to variants in the primer-binding regions, and (b) the presence of multiple indels in female carriers which confounded analysis and interpretation.

## 4. Discussion

In this study, we identified pathogenic *RPGR* ORF15 variants in 13% (*n* = 11/82 probands) of an African IRD cohort comprising isolated affected males and families lacking male-to-male disease transmission. A total of 32 unique variants were identified (18 validated by Sanger sequencing), including novel pathogenic frameshift mutations.

The prevalence of small indels in *RPGR* exon ORF15 (57 in 82 samples sequenced in this cohort) indicate that the replication of this region in vivo is demanding and error-prone. This might explain, at least partly, the challenge of amplification in vitro. In fact, PCR amplification failed several times, for both NGS (*n* = 15/97 samples; 15% of the cohort) and Sanger protocols (*n* = 5/33 samples; 15% of the cohort). On the other hand, it is possible that variants in the primer-binding regions prevent successful amplification. Most publications only report “positive results” and do not comment on amplification failure. ORF15 is a known troublesome region, and it would be valuable to know what proportion of ORF15 amplifications fail, in particular for older DNA samples, in other laboratories.

Heterozygous variant calls resulting from the standard secondary analysis pipeline (BWA-MEM and GATK) in male samples highlighted problems that occurred during alignment and/or variant calling across this low-complexity region. Furthermore, we identified 516 potential false positives produced by the standard pipeline, which passed all quality filters (85% of all variant calls). We were able to circumvent the problem by developing an alternative secondary analysis pipeline based on de novo assembly. This analysis allowed for the identification of otherwise missed variants, in particular indels larger than 15 bp. By comparing the results of both data analysis pipelines, we showed that the novel method is more sensitive (in particular for indels) and less prone to false positive calls, as illustrated in Figure 1. The novel pipeline produced a maximum of 67 potential false positive calls, 47 of which are accounted for by a single variant (c.2863T>G). The results seem to suggest superiority of the newly developed analysis method, in terms of sensitivity and specificity.

However, the continued identification of false positives in the novel pipeline’s output warrants caution in results interpretation. All of these (except for c.2820A>G) were present in the standard pipeline’s output, too. Furthermore, the unavailability of verified true heterozygous variants in female probands (16 included in this study) made it impossible to verify how the pipeline will perform in the presence of two alleles. Sanger sequencing ORF15 in females can be very complicated due to the high frequency of small indels (often heterozygous in females), which shift the frame of the Sanger traces and can hinder its analysis. Moreover, this problem affects all other current methodologies in the case of ORF15. Based on this, we recommend the parallel use of both pipelines for the analysis of *RPGR*’s ORF15 by NGS and to perform the analysis on affected males where possible.

Our analysis of the African cohort identified a large number (*n* = 15/32 as listed on Table 1; 47%) of variants not represented in gnomAD and lacking an rs ID, which we have to presume are novel or rare. The count is inflated by variants from samples Afr-61 and Afr-31; as discussed above, Afr-61 is reported to carry three novel variants, but the more conservative variant calls described above would reduce it to one novel and one very rare variant. Afr-31 contributes four additional novel variants, two of which were not present on Sanger sequences, whilst the remaining two could not be verified. Therefore, a more conservative count lists 10 novel variants out of 28 unique variants (35%). However, it is possible that these variants have been missed by gnomAD, and others, due to complications with the capture or alignment of this genomic region, and this limitation should be noted. Nevertheless, this work once again underscores the genetic diversity of the understudied African populations and highlights their value in molecular studies by (1) contributing towards a more comprehensive global catalogue of disease-causing mutations and (2) providing additional information for pathogenicity interpretation. For example, the ‘novel’ variants of uncertain clinical significance c.2606_2632dupAAGAAGGGGAGGAAGGAGAAGGGGAGG and c.2618_2632dupAAGGAGAAGGGGAGG are relatively common in the indigenous African study cohort (9% and 6% of the cohort, respectively), providing supporting evidence that they are likely benign.

Two novel pathogenic frameshift mutations, namely c.2457_2460delAGAG and c.2470_2471delGG, were identified. The latter mutation was recurrent in two families, Afr-63 and Afr-65, both of unknown ethnolinguistic origin. Similarly, the c.2790_2791delGG pathogenic variant was identified in two additional families in this cohort, bringing the total to four indigenous African families of different ethnolinguistic groups; two previously identified large Xhosa families (Appendix A), as well as Afr-16 (ethnicity was not reported) and Afr-71 (self-identified as Tswana) in this cohort (Figure 2). The Sotho-Tswana and Xhosa groups show significant genetic differences, despite recent geographic and linguistic divergence [30]. Although we cannot exclude gene flow after divergence, c.2790_2791delGG is possibly an ancient mutation from a common ancestor and could be relatively prevalent in SA. In addition, a third recurrent pathogenic mutation was identified, namely c.2964_2965delGG (in self-reported Zulu/Swazi individual Afr-60, and two Xhosa families i.e., Afr-75, and the previously identified family in Appendix A). These three recurrent mutations (c.2470_2471delGG, c.2790_2791delGG and c.2964_2965delGG) provide a possible target for an initial population-relevant, cost-effective diagnostic screen, as combined they account for 9/14 (64%) of ORF15-associated indigenous African families identified to date.

Phenotypically, all probands identified with pathogenic mutations manifested with early-onset RP. The exception was Afr-38, which was diagnosed with rod-cone macular degeneration and a self-reported onset age of 36. Whilst cone involvement is not unusual at an advanced stage of RP, this onset-age outlier is likely due to a combination of factors such as language barriers and limited access to ophthalmological services in SA.

Approximately half of the resolved African families showed manifesting female carriers (Figure 2). In SA’s resource-limited environment, special consideration should therefore be given to pre-screening ORF15 in apparently dominant pedigrees lacking male-to-male transmission and exhibiting early disease onset, prior to costly WES, which would have almost certainly missed the causal variants. Expanding on this approach, the 71 probands that have been excluded from harbouring causal ORF15 mutations are now eligible for NGS screening (either via WES or targeted capture of remaining candidate genes), in the hopes of finding the genetic causes of blindness.

Finally, the genetic diagnosis achieved by our new sequence data analysis pipeline will impact the care and counselling for these 11 patients and their extended families. Identification of the underlying mutations allows for diagnostic closure and enables calculation of the risk of developing the disease, thereby facilitating decision making and life choices. Importantly, a molecular diagnosis may also determine eligibility for a potentially available gene-based and personalised therapy (https://clinicaltrials.gov/ct2/show/NCT03116113) and/or other treatments and interventions in order to preserve vision.

## Figures and Tables

**Figure 1 genes-11-00800-f001:**
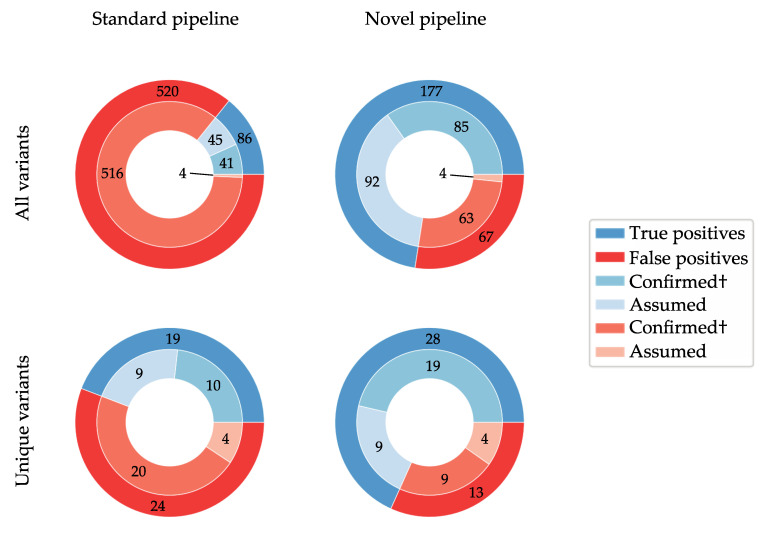
Nested pie plots highlighting the proportion of true versus false positive variant calls produced by the standard and novel pipelines. Blue shaded portions represent true positives; red portions illustrate false positive calls. The top charts summarise all variant calls (*n* = 606 and *n* = 244 variant calls in standard and novel pipelines, respectively), and the bottom charts represent all unique variants (*n* = 43 and *n* = 41 variants, respectively). The inner rings show the proportion of each category that has been verified by Sanger sequencing (Confirmed†) and those that could not be confirmed (Assumed). As listed in Table 1 and Table 2, the Confirmed† categories include all variants that have been confirmed in at least a subset of samples.

**Figure 2 genes-11-00800-f002:**
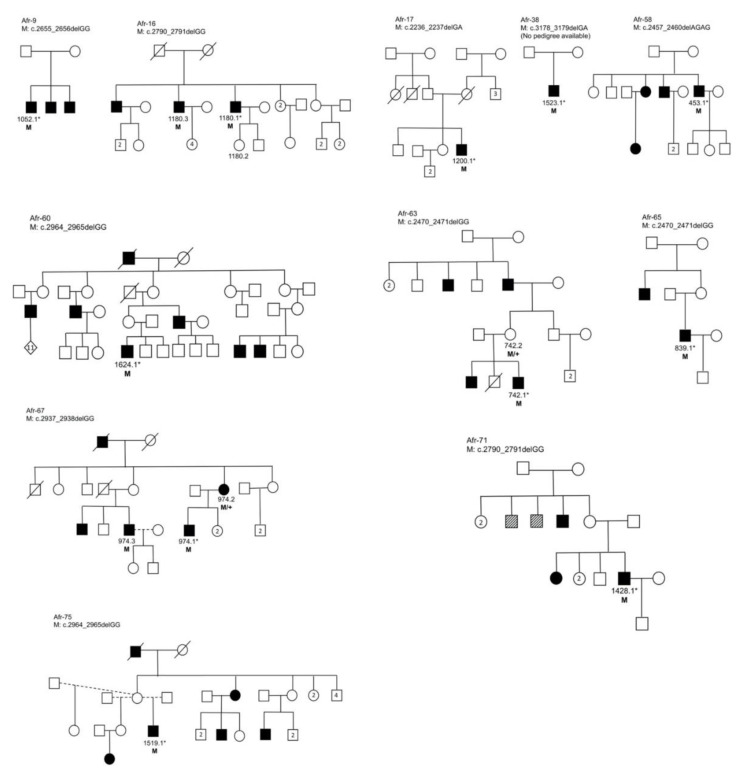
Pedigrees of 11 families with verified pathogenic open reading frame 15 (ORF15) mutations. Squares represent males, and circles, females. Shaded symbols indicate individuals affected with inherited retinal disorders (IRDs), and hatched symbols indicate individuals with undiagnosed visual problems. Identifier codes show individuals from whom biological material was available, and those selected for next-generation sequencing (NGS) are noted with an asterisk. Segregation of the mutation in the families is indicated as: M (hemizygous mutation); M/+ (heterozygous mutation).

**Table 1 genes-11-00800-t001:** Spectrum of variants identified in the cohort by the novel pipeline. Variant nomenclature based on hg19 with the corresponding gnomAD overall frequency (when available) and the number of samples in the cohort carrying each variant. Column “Std pipeline” displays whether the variant was identified by the standard pipeline or not. The “Sanger” column shows whether the variant was identified in the available sequences. Variants highlighted in grey and in italics are confirmed false positives. Bold orange variants were considered pathogenic. * denotes variants that have been flagged as low quality on gnomAD. † denotes variants for which only a subset of samples was verified by Sanger.

cNomen (NM_001034853.2)	pNomen	# of Samples	gnomAD (%)	rsID	Pathogenicity	Std Pipeline	Sanger	First Report
c.*125delA		52	31.2591	rs35637775	Benign	No		
c.3407G>A	p.Gly1136Asp	2	0.2171	rs150960964	Benign	Yes		
c.3396C>T	p.=	4	16.1297	rs12687163	Benign	Yes		
c.3264G>A	p.=	14	7.7524	rs78736275	Benign	Yes		
c.3219C>T	p.=	14	7.7934	rs111787313	Benign	Yes	Yes †	
**c.3178_3179del**	**p.Glu1060ArgfsTer18**	**1**	**0.0011**	**rs771214648**	**Pathogenic**	**Yes**	**Yes**	**1**
c.3108_3122del	p.Glu1038_Glu1042del	1	0.2447	rs774012136	Benign	Yes		
*c.3074T>G*	*p.Val1025Gly*	*1*	*0.0169*	*rs773842474*	*Likely benign*	*Yes*	*No*	
*c.3062T>G*	*p.Val1021Gly*	*1*	*0.8368**	*rs144299434*	*Likely benign*	*Yes*	*No*	
**c.2964_2965del**	**p.Glu989GlyfsTer89**	**2**	**NA**		**Pathogenic**	**Yes**	**Yes**	**1**
c.2919_2939dup	p.Gly977_Glu983dup	13	1.0322*	rs772859148	VUS	No	Yes †	
**c.2937_2938del**	**p.Glu980GlyfsTer98**	**1**	**NA**		**Pathogenic**	**Yes**	**Yes**	**2**
*c.2863T>G*	*p.Trp955Gly*	*47*	*19.7002*	*rs765224300*	*Benign*	*Yes*	*No †*	
c.2820_2840dup	p.Asp943_Glu949dup	5	3.3006	rs764268405	Benign	No	Yes	
c.2829T>G	p.Asp943Glu	1	1.4892	rs201655057	Likely benign	Yes		
c.2829T>A	p.Asp943Glu	1	0.133	rs201655057	Likely benign	Yes		
*c.2820A>G*	*p.=*	*1*	*NA*		*VUS*	*No*	*No*	
**c.2790_2791del**	**p.Glu931GlyfsTer147**	**2**	**NA**		**Pathogenic**	**Yes**	**Yes**	**3**
c.2778_2786del	p.Glu927_Glu929del	1	NA		VUS	Yes		
*c.2758A>G*	*p.Lys920Glu*	*4*	*0.0548*	*rs1241379586*	*Likely benign*	*Yes*	*No †*	
c.2705G>A	p.Gly902Glu	1	NA		Likely benign	Yes		
c.2667_2669del	p.Glu890del	1	1.6129	rs1263452259	Benign	Yes		
c.2660_2661insGGAAGAGGAGGAAGGAGAAGGGGAGGGAGAAGAGGAAGGAGAAGGGGAGGG	p.Glu890_Gly891insGluGlyGluGlyGluGlyGluGluGluGlyGluGlyGluGlyGluGluGlu	1	NA		VUS	No	Yes	
**c.2655_2656del**	**p.Glu886GlyfsTer192**	**1**	**NA**		**Pathogenic**	**Yes**	**Yes**	**4**
*c.2639A>G*	*p.Glu880Gly*	*1*	*NA*		*Likely benign*	*Yes*	*Yes*	
c.2633_2634delinsAA	p.Gly878Glu	1	NA		Likely benign	Yes	Yes	
c.2618_2632dup	p.Glu873_Glu877dup	5	NA		VUS	No	Yes †	
c.2606_2632dup	p.Glu869_Glu877dup	6	NA	rs769216492	VUS	No	Yes †	
*c.2569A>G*	*p.Lys857Glu*	*6*	*0.981*	*rs1250133030*	*Likely benign*	*Yes*	*No †*	
c.2541_2561del	p.Glu850_Gly856del	12	2.0962	rs886038384	Likely benign	No	Yes †	
c.2514G>A	p.=	1	NA		VUS	Yes		
*c.2499T>G*	*p.=*	*1*	*0.0406*	*rs752979508*	*Likely benign*	*Yes*	*No*	
**c.2470_2471del**	**p.Gly824ArgfsTer10**	**2**	**NA**		**Pathogenic**	**Yes**	**Yes**	**This study**
**c.2457_2460del**	**p.Glu820ArgfsTer268**	**1**	**NA**		**Pathogenic**	**Yes**	**Yes**	**This study**
c.2341G>A	p.Ala781Thr	2	12.1386	rs5917557	Benign	Yes		
**c.2236_2237del**	**p.Glu746ArgfsTer23**	**1**	**NA**		**Pathogenic**	**Yes**	**Yes**	**5**
c.2223G>A	p.=	15	10.3568	rs147619484	Benign	Yes	Yes †	
c.2057T>A	p.Met686Lys	1	0.449	rs151247357	Likely benign	Yes		
*c.1961G>A*	*p.Arg654Lys*	*1*	*NA*		*Likely benign*	*Yes*	*No*	
*c.1933G>A*	*p.Gly645Arg*	*1*	*NA*		*Likely benign*	*Yes*	*No*	
c.1754-103C>T		15	10.1753	rs41303691	Benign	Yes		

**Table 2 genes-11-00800-t002:** Standard secondary analysis pipeline variants passing quality filters. Column four highlights how often the variant had been called as heterozygous and how many of these were from male samples. Column six shows whether the variant had been identified by the novel pipeline and in how many samples. Sanger sequences were used to confirm the presence or absence of the variant. Variants highlighted in grey, and in italics are confirmed false positives. Bold orange variants were considered pathogenic. † denotes variants for which only a subset of samples was verified by Sanger.

cNomen (NM_001034853.2)	pNomen	# of Samples	Het. (Het. Males)	Hom./Hemi.	Novel Pipeline (# of Samples)	Call Quality	Sanger
c.3407G>A	p.Gly1136Asp	3	2 (0)	1	Yes (2)	High	
c.3396C>T	p.=	5	2 (0)	3	Yes (4)	High	
c.3356G>A	p.Arg1119Lys	1	1 (1)	0	No	Low	
c.3264G>A	p.=	15	4 (0)	11	Yes (14)	High	
c.3219C>T	p.=	15	4 (0)	11	Yes (14)	High	Yes †
**c.3178_3179del**	**p.Glu1060ArgfsTer18**	**1**	**0 (0)**	**1**	**Yes (1)**	**High**	**Yes**
c.3108_3122del	p.Glu1038_Glu1042del	1	0 (0)	1	Yes (1)	High	
*c.3074T>G*	*p.Val1025Gly*	*80*	*80 (64)*	*0*	*Yes (1)*	*Low*	*No †*
*c.3062T>G*	*p.Val1021Gly*	*74*	*74 (58)*	*0*	*Yes (1)*	*Low*	*No †*
**c.2964_2965del**	**p.Glu989GlyfsTer89**	**2**	**2 (2)**	**0**	**Yes (2)**	**High**	**Yes**
**c.2937_2938del**	**p.Glu980GlyfsTer98**	**1**	**1 (1)**	**0**	**Yes (1)**	**High**	**Yes**
*c.2895G>A*	*p.=*	*8*	*8 (7)*	*0*	*No*	*Low*	*No †*
*c.2876A>G*	*p.Glu959Gly*	*1*	*1 (1)*	*0*	*No*	*Low*	*No*
*c.2863T>G*	*p.Trp955Gly*	*81*	*81 (65)*	*0*	*Yes (47)*	*Low*	*No †*
*c.2847A>G*	*p.=*	*1*	*1 (1)*	*0*	*No*	*Low*	*No*
*c.2829T>G*	*p.Asp943Glu*	*50*	*50 (44)*	*0*	*Yes (1)*	*Low*	*No †*
*c.2829T>A*	*p.Asp943Glu*	*30*	*40 (23)*	*0*	*Yes (1)*	*Low*	*No †*
**c.2790_2791del**	**p.Glu931GlyfsTer147**	**2**	**2 (2)**	**0**	**Yes (2)**	**High**	**Yes**
c.2778_2786del	p.Glu927_Glu929del	1	0 (0)	1	Yes (1)	High	
*c.2784A>G*	*p.=*	*1*	*1 (1)*	*0*	*No*	*Low*	*No*
*c.2758A>G*	*p.Lys920Glu*	*75*	*75 (59)*	*0*	*Yes (4)*	*Low*	*No †*
c.2705G>A	p.Gly902Glu	1	1 (1)	0	Yes (1)	High	
c.2667_2669del	p.Glu890del	2	1 (0)	1	Yes (1)	High	
**c.2655_2656del**	**p.Glu886GlyfsTer192**	**1**	**1 (1)**	**0**	**Yes (1)**	**High**	**Yes**
*c.2639A>G*	*p.Glu880Gly*	*6*	*6 (6)*	*0*	*Yes (1)*	*Low*	*No †*
*c.2634G>A*	*p.=*	*6*	*6 (6)*	*0*	*Yes (1)*	*Low*	*No †*
*c.2633G>A*	*p.Gly878Glu*	*6*	*6 (6)*	*0*	*Yes (1)*	*Low*	*No †*
*c.2589A>G*	*p.=*	*7*	*7 (6)*	*0*	*No*	*Low*	*No †*
*c.2569A>G*	*p.Lys857Glu*	*11*	*11 (9)*	*0*	*Yes (6)*	*Low*	*No †*
c.2531A>G	p.Glu844Gly	1	1 (0)	0	No	Low	
*c.2517A>G*	*p.=*	*11*	*11 (10)*	*0*	*No*	*Low*	*No †*
c.2514G>A	p.=	1	1 (1)	0	Yes (1)	High	
*c.2499T>G*	*p.=*	*65*	*65 (54)*	*0*	*Yes (1)*	*Low*	*No †*
**c.2470_2471del**	**p.Gly824ArgfsTer10**	**2**	**0 (0)**	**2**	**Yes (2)**	**High**	**Yes**
**c.2457_2460del**	**p.Glu820ArgfsTer268**	**1**	**0 (0)**	**1**	**Yes (1)**	**High**	**Yes**
c.2341G>A	p.Ala781Thr	2	1 (0)	1	Yes (2)	High	
**c.2236_2237del**	**p.Glu746ArgfsTer23**	**1**	**0 (0)**	**1**	**Yes (1)**	**High**	**Yes**
c.2223G>A	p.=	15	3 (0)	12	Yes (15)	High	Yes †
c.2057T>A	p.Met686Lys	1	0 (0)	1	Yes (1)	High	
*c.1961G>A*	*p.Arg654Lys*	*1*	*1 (1)*	*0*	*Yes (1)*	*High*	*No*
*c.1933G>A*	*p.Gly645Arg*	*1*	*1 (1)*	*0*	*Yes (1)*	*High*	*No*
*c.1885G>A*	*p.Asp629Asn*	*1*	*1 (1)*	*0*	*No*	*Low*	*No*
c.1754-103C>T		15	10.1753	rs41303691	Benign	Yes

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
