# Peer review of "De Novo Assembly-Based Analysis of RPGR Exon ORF15 in an Indigenous African Cohort Overcomes Limitations of a Standard Next-Generation Sequencing (NGS) Data Analysis Pipeline"

_genes, 2020, doi:10.3390/genes11070800_

Round 1

Reviewer 1 Report

Variants in RPGR exon ORF15 are one of the most common causes of X-linked RP. The AG-rich and the low sequence complexity within this region makes it not only challenging for sequencing data generation but also for subsequent data analysis.  In this paper, the authors developed an alternative Python-based pipeline integrating the de novo assembly software SPAdes, the multi-fasta aligner 199 MAFFT v7, and the SNVs identified SNP-sites to improve the molecular diagnostic for RPGR ORF15. The paper showed that the method has improved sensitivity over the standard NGS pipeline that allows for the identification of otherwise missed variants. The authors also showed that the method is less prone to false-positive calls, avoiding the miscalling of variants generated by standard NGS analysis. The results suggest that the new analysis tool is superior in terms of sensitivity and specificity.

The method is not perfect. The authors still identified false positives which all of these were also present in the standard NGS output. Moreover, the authors are not sure if the new pipeline will work for heterozygous alleles due to the unavailability of verified heterozygous female probands in their study. The paper discussed these shortcomings and suggested proper ways to obtain a more accurate diagnosis.

The paper is very well written. The method developed is clearly an improvement for care and counseling for RPGR ORF15 patients and their families, in terms of decision-making and life choices, and for determining eligibility for potentially available gene therapy.

Author Response

Point 1: Variants in RPGR exon ORF15 are one of the most common causes of X-linked RP. The AG-rich and the low sequence complexity within this region makes it not only challenging for sequencing data generation but also for subsequent data analysis.  In this paper, the authors developed an alternative Python-based pipeline integrating the de novo assembly software SPAdes, the multi-fasta aligner 199 MAFFT v7, and the SNVs identified SNP-sites to improve the molecular diagnostic for RPGR ORF15. The paper showed that the method has improved sensitivity over the standard NGS pipeline that allows for the identification of otherwise missed variants. The authors also showed that the method is less prone to false-positive calls, avoiding the miscalling of variants generated by standard NGS analysis. The results suggest that the new analysis tool is superior in terms of sensitivity and specificity.

The method is not perfect. The authors still identified false positives which all of these were also present in the standard NGS output. Moreover, the authors are not sure if the new pipeline will work for heterozygous alleles due to the unavailability of verified heterozygous female probands in their study. The paper discussed these shortcomings and suggested proper ways to obtain a more accurate diagnosis.

The paper is very well written. The method developed is clearly an improvement for care and counseling for RPGR ORF15 patients and their families, in terms of decision-making and life choices, and for determining eligibility for potentially available gene therapy.

Response 1:

We thank reviewer 1 for critically reading our manuscript and the positive feedback. We acknowledge that the method is not perfect and we pointed out all of the shortcomings that we were able to identify with the datasets analysed during this study. Despite it not being perfect, we think it is an improvement compared to the standard pipeline that most labs use to date.

We regret that we were unable to verify heterozygosity in females and see how the novel analysis tool behaves in this situation. By comparing the standard and novel pipeline results, however, we noticed that for some high quality variants, the standard pipeline had additional heterozygous calls for female samples that were missing in the novel pipeline’s output. We think this is evidence of the fact that the novel pipeline might miss heterozygous calls in females. The data seems to suggest that the variant is integrated in the contig in roughly 50% of the cases. As explained in the manuscript, we were unable to verify this by Sanger sequencing. Sanger sequencing ORF15 in females can be very complicated due to the high frequency of small indels (often heterozygous in females), which shift the frame of the Sanger traces and can hinder its analysis. Moreover, this problem affects all other current methodology in the case of ORF15.

However, xlRP affects mostly males and generally analysis would start with sequencing of a male index patient.

We have added a comment to this effect, on lines 346-349:

“Sanger sequencing ORF15 in females can be very complicated due to the high frequency of small indels (often heterozygous in females), which shift the frame of the Sanger traces and can hinder its analysis. Moreover, this problem affects all other current methodology in the case of ORF15.”

Reviewer 2 Report

I think that overall the paper is clearly written and addresses a topic of general interest. It also describes a small number of novel pathogenic variants. Some minor typological corrections:

Line 47 – maybe better to clarify that the most severe RP phenotypes in males are caused by ORF15.

Line 64 – starts with “Another limitation of NGS” – however this does not really read correctly as it is not preceeded by a limitation of NGS. It might better read “One limitation of NGS….”.

Line 244 – Call needs to be plural

Table 2 – Correct p.Afr1119Lys on third line

In table 2 you have a column labelled hom – are some of these hemi?

However in general I am concerned at the clarity with which the results are presented and the amount of missing data (15% of NGS and 15% of Sanger). I acknowledge the authors honesty in this regard. 

It is also of concern that the authors are unable to verify heterozygous females in their proposed pipeline. 

One specific point is that false calls are presented in the same tables as real calls and it would take an in depth read to distinguish the two - a cursory reading would leave the reader open to missenterpretation as pathogenic variants are presented alongside misscalled ones.

There is a lot of data spread over two large tables - this format needs to be reviewed critically as it is very difficult to read and interpret at the moment.

It is confusing to describe heterozygous variants in male samples. 

Author Response

Point 1: I think that overall the paper is clearly written and addresses a topic of general interest. It also describes a small number of novel pathogenic variants. Some minor typological corrections:

Line 47 – maybe better to clarify that the most severe RP phenotypes in males are caused by ORF15.

Line 64 – starts with “Another limitation of NGS” – however this does not really read correctly as it is not preceeded by a limitation of NGS. It might better read “One limitation of NGS….”.

Line 244 – Call needs to be plural

Table 2 – Correct p.Afr1119Lys on third line

Response 1:

We thank reviewer 2 for critically reading our manuscript and the constructive feedback.

We addressed the minor typological changes on lines 47, 64, 245, and Table 2.

Point 2: In table 2 you have a column labelled hom – are some of these hemi?

Response 2:

Yes, most of them actually are hemizygous. We edited the column’s heading on Table 2 to reflect that.

Point 3: However in general I am concerned at the clarity with which the results are presented and the amount of missing data (15% of NGS and 15% of Sanger). I acknowledge the authors honesty in this regard. 

Response 3:

We acknowledge the fact that it is concerning that such a high proportion of samples could not be amplified by PCR. However, we feel like this is nothing unusual. Most published articles only publish “positive results” and do not report on PCRs could not be performed. ORF15 is a known troublesome region and we can imagine other labs having similar problems with its amplification, in particular for older DNA samples. It would be interesting to know the portion of samples for which ORF15 PCR does not work in other labs.

We have added a comment to this effect on line 314:

“Most publications only report “positive results” and do not comment on amplification failure. ORF15 is a known troublesome region and it would be valuable to know what proportion of ORF15 amplifications fail, in particular for older DNA samples, in other laboratories.”

Point 4: It is also of concern that the authors are unable to verify heterozygous females in their proposed pipeline.

Response 4:

We regret that we were unable to verify heterozygosity in females and see how the novel analysis tool behaves in this situation. By comparing the standard and novel pipeline results, however, we noticed that for some high quality variants, the standard pipeline had additional heterozygous calls for female samples that were missing in the novel pipeline’s output. We think this is evidence of the fact that the novel pipeline might miss heterozygous calls in females. The data seems to suggest that the variant is integrated in the contig in roughly 50% of the cases. As explained in the manuscript, we were unable to verify this by Sanger sequencing. Sanger sequencing ORF15 in females can be very complicated due to the high frequency of small indels (often heterozygous in females), which shift the frame of the Sanger traces and can hinder its analysis. Moreover, this problem affects all other current methodology in the case of ORF15.

However, xlRP affects mostly males and generally analysis would start with sequencing of a male index patient.

We have added a comment to this effect, on lines 346-349:

“Sanger sequencing ORF15 in females can be very complicated due to the high frequency of small indels (often heterozygous in females), which shift the frame of the Sanger traces and can hinder its analysis. Moreover, this problem affects all other current methodology in the case of ORF15.”

Point 5 & 6: One specific point is that false calls are presented in the same tables as real calls and it would take an in depth read to distinguish the two - a cursory reading would leave the reader open to missenterpretation as pathogenic variants are presented alongside misscalled ones.

There is a lot of data spread over two large tables - this format needs to be reviewed critically as it is very difficult to read and interpret at the moment.

Response 5 & 6:

We agree that the tables are rich in information. We wanted to present the results this way to reflect the outputs of both pipelines since we discuss them in detail. However, we agree that the false positives should be more readily identifiable. Therefore, we have highlighted the pathogenic variants in orange with bold font, whilst we highlighted the false positive variants in grey and italic font. This modification should improve the readability of the tables. We edited the Tables’ legends to reflect these changes on lines 235-236 and 286-287):

“Variants highlighted in grey and in italics are confirmed false positives. Bold orange variants were considered pathogenic.”

To give a more visual representation of the overall performance of each pipeline, we have created an additional figure (Figure 1) that illustrates the portions of true versus false positives in either pipelines outputs (lines 290-303). We think this will help with overall interpretation of the results.

We added a small paragraph to introduce Figure 1 on lines 290-294:

“Figure 1 visually summarizes the findings contained in Table 1 and 2 by illustrating the numbers of overall and unique true and false positives identified by each pipeline. It is important to point out once more that each unique variant call, which is present in multiple samples, has been Sanger verified in a subset of those. However, since all of them are concordant, all such variant calls have been considered as “confirmed” false or true positives in Figure 1.”

Point 7: It is confusing to describe heterozygous variants in male samples. 

Response 7:

We agree, and it is wrong. However, that is the standard pipeline’s output for many variants passing all quality filters, which is concerning and prompted us to develop an alternative pipeline in the first place.

We edited lines 192 to stress this issue:

“Specifically, it often resulted in erroneous “heterozygous” calls in males, and was particularly poor at detecting larger indels (≥ 20bp).”

Round 2

Reviewer 2 Report

The authors have addressed all my comments/suggestions to my satisfaction.